# Medicinal Components in Edible Mushrooms on Diabetes Mellitus Treatment

**DOI:** 10.3390/pharmaceutics14020436

**Published:** 2022-02-17

**Authors:** Arpita Das, Chiao-Ming Chen, Shu-Chi Mu, Shu-Hui Yang, Yu-Ming Ju, Sing-Chung Li

**Affiliations:** 1School of Nutrition and Health Sciences, College of Nutrition, Taipei Medical University, Taipei 11031, Taiwan; da07110008@tmu.edu.tw; 2Department of Food Science, Nutrition and Nutraceutical Biotechnology, Shih Chien University, Taipei 10462, Taiwan; charming@g2.usc.edu.tw; 3Department of Pediatrics, Shin-Kong Wu Ho-Su Memorial Hospital, Taipei 11101, Taiwan; musc1004@gmail.com; 4School of Medicine, College of Medicine, Fu-Jen Catholic University, New Taipei City 24205, Taiwan; 5Fengshan Tropical Horticultural Experiment Branch, Taiwan Agricultural Research Institute, Kaohsiung City 83052, Taiwan; debbie@tari.gov.tw; 6Institute of Plant and Microbial Biology, Academia Sinica, Taipei 11529, Taiwan; yumingju@gate.sinica.edu.tw

**Keywords:** edible mushroom, insulin resistance, diabetes, polysaccharide, vitamin D, α-glucosidase, α-amylase

## Abstract

Mushrooms belong to the family “Fungi” and became famous for their medicinal properties and easy accessibility all over the world. Because of its pharmaceutical properties, including anti-diabetic, anti-inflammatory, anti-cancer, and antioxidant properties, it became a hot topic among scientists. However, depending on species and varieties, most of the medicinal properties became indistinct. With this interest, an attempt has been made to scrutinize the role of edible mushrooms (EM) in diabetes mellitus treatment. A systematic contemporary literature review has been carried out from all records such as Science Direct, PubMed, Embase, and Google Scholar with an aim to represents the work has performed on mushrooms focuses on diabetes, insulin resistance (IR), and preventive mechanism of IR, using different kinds of mushroom extracts. The final review represents that EM plays an important role in anticipation of insulin resistance with the help of active compounds, i.e., polysaccharide, vitamin D, and signifies α-glucosidase or α-amylase preventive activities. Although most of the mechanism is not clear yet, many varieties of mushrooms’ medicinal properties have not been studied properly. So, in the future, further investigation is needed on edible medicinal mushrooms to overcome the research gap to use its clinical potential to prevent non-communicable diseases.

## 1. Introduction

The word “mushroom” is derived from the Latin and Greek words “fungus” and “mykes” and is considered edible if consumption does not cause health disorders. Based on the edibility, it can be divided into three groups-edible (*Lepiota procera*), inedible or poisonous (*Lepiota margani*), and non-poisonous [1]. For thousands of years, mushrooms have been used as food and medicine (38%), and many Asian and south-Asian countries use traditional wild edible mushrooms as appealing and nutritious foods, including China, Japan, India, and Taiwan. As a result, mushrooms received a high demand in the global market with 34 billion kg production and per capita consumption of 4.7 kg in 2013. Some of the significant ordinarily available edible mushrooms are chanterelles (*Cantharellaceae*), puffballs (*Lycoperdon* spp. and *Calvatia* spp.), shaggy mane (*Coprinus comatus*), two oyster mushrooms (*Pleurotus ostreatus* and *Pleurotus cystidiosus*), boletes (*Boletaceae*), sulfer shelf (*Laetiporus sulphurous*), hen of the woods (*Grifora frondosa*), button mushroom (*Agricus bisporus*), golden oyster mushroom (*Pleurotus citrinopileatus*) [2,3,4], morels (*Morachella esculenta*), bearded tooth (*Hericium erinaceus*), straw mushroom (*Volvariella volvacea*) [5,6], Eenoki (*Flammulina velutipes*) [7], shiitake (*Lentinula edodes*) [8], beech mushroom (*Hypsizygus marmoreus*) [9], french horn mushroom (*Pleurotus eryngii*) [10], dancing mushroom (*Grifola frondosa*) [11], and black poplar mushroom (*Agrocybe aegerita*) [12]. Wild edible mushrooms are not only superior for the chemical and nutritional characteristics but also the protein and vitamin contents, B vitamins, vitamin D, vitamin K, and rarely vitamin A and C [13,14,15]. Moreover, mushrooms have low-fat content and are high in dietary fiber, nutraceuticals, and polysaccharides, which show positive health benefits on several diseases through their immunomodulatory and anti-inflammation properties [16,17]. The core structural polysaccharides from mushrooms include homoglucans (*ß*-1, 3 glucan), heteroglycans, heterogalactans, and heteromannans. These polysaccharides and terpenoids (secondary metabolites) play a pivotal role in glucose homeostasis by inhibiting α-glucosidase, assisting glucose transporter 4 actions, and reducing inflammatory factors to improve insulin resistance and lipid metabolism [18,19].

The term “diabetes mellitus” or “DM” is derived from the Greek word “diabetes”, which means siphon, to pass through, and the Latin word “mellitus”, which means sweet. Diabetes mellitus is a group of non-communicable metabolic diseases characterized by prolonged hyperglycemia resulting from defects in insulin secretion, insulin action, or both [20]. According to American Diabetes Association, 1997 (ADA), diabetes mellitus is classified as type 1 diabetes mellitus (T1DM) (insulin-dependent or juvenile-onset; accounting for 3–10% cases), type 2 diabetes mellitus (T2DM) (non-insulin-dependent or adult-onset; accounting 85–90% cases), and gestational diabetes mellitus (hyperglycemia occurs during the second or third trimester of pregnancy and generally resolves after delivery; accounting 2–5% cases) [21,22]. According to the International Diabetes Federation Data in 2015, 415 million people (80% from middle- and low-income family) was suffering from diabetes, and if it continues to grow will reach 642 million people by 2040 [23]. T1DM is generally accompanied by an autoimmune disorder that triggers the destruction of pancreatic beta cells, alterations in lipid metabolism, enhanced hyperglycemia-mediated oxidative stress, endothelial cell dysfunction, and apoptosis [24,25]. Whereas T2DM causes glucotoxicity, lipotoxicity, endoplasmic reticulum-induced stress, and apoptosis, which finally leads to progressive loss of beta cells [26]. Specific symptom includes polydipsia, polyphagia, polyuria, and nocturia, whereas complication comprises microvascular (retinopathy, nephropathy, and neuropathy) and macrovascular (ischemic heart disease, peripheral vascular disease, and cerebrovascular disease) abnormalities. Several studies have proven that insulin resistance (IR) is the main factor to be concerned about in complications of DM [26,27,28]. Besides IR, some of the common risk factors for insulin resistance are oxidative stress, hydrolytic enzymatic inhibition, inflammation, genetic habitual, environmental, dyslipidemia, obesity, and epigenetic modulations [29,30]. Thus, many pathological factors use to contribute to insulin resistance, although the exact mechanism is not clear yet.

By considering the following factors, the main aim of this review is to elucidate the role of edible mushroom (EM) in diabetes mellitus (DM) treatment by monitoring bioactive compounds of mushrooms, pathophysiology of insulin resistance (IR), and the preventive mechanism of IR using EMs. The data were retrieved by searching scientific publications (research and/or review papers) from databases including Science Direct, PubMed, Embase, and Google Scholar with keywords such as “diabetes”, “insulin resistance”, “mushroom extracts”, “in vivo and in vitro studies”, etc. A total of 100 publications were collected, including in vivo and in vitro studies (Figure 1), in which 23 common edible mushroom varieties have been identified (Table 1), and further discussion has been carried out based on scientific literature availability on DM.

We further investigated different species of edible mushrooms and presented the data based on hypoglycemic compounds. After considering the potential bioactive compounds, in vivo and in vitro research analysis was carried out, and the information is represented in Table 2.

## 2. Diabetes Mellitus and Insulin Resistance

Nowadays, the global incidence of diabetes is escalating in both developed and developing countries. Generally, obesity is directly involved in the pathogenesis of T2DM along with insulin resistance (IR), and the incidence of IR in T1DM is also increasing frequently. Therefore, it is crucial to understand the mechanism of glucose homeostasis and insulin resistance.

Pancreatic cell plays an important role in glucose homeostasis. T2DM reduces insulin secretion by 50% and also lessens the sensitivity of peripheral tissue to insulin up to 70%. So, the study on IR has a great clinical significance in medical interventions. Basically, insulin exerts its effect through phosphorylation of phosphoinositide 3-kinase (PI3K) and protein kinase B (PkB, Akt) [72,73]. Next, through PI3k phosphorylation, it activates glucose transporter 4 (GLUT4). Other factors involved in insulin action and carbohydrate metabolism are serine/threonine kinase Akt phosphorylates GSk3 *ß*, FOXOs, IRS (insulin-regulated sequence), and SREBP-1c (sterol-regulated element-binding protein transcription factors) (Figure 2) [74].

Insulin sensitivity can be defined as a pathological condition in which normal plasma insulin level in targeted tissues fails to maintain normal blood glucose levels via enhancement of endogenous glucose production, lipogenesis, and glycolysis [75]. Thereby, IR refers to a state that exhibits the reduced biological effect of a given insulin concentration. So, IR increases insulin secretion and thus enhances fasting plasma insulin level, which is finally considered IR [76]. Moreover, IR is not only associated with DM but also several pathological conditions such as cardiovascular disease, non-alcoholic fatty liver disease, and cancer as well [77].

## 3. Diabetes Mellitus and Insulin Resistance Preventive Mechanism by Edible Mushroom

Mushrooms possess medicinal components due to the presence of different types of secondary metabolites such as polysaccharides, lectins, lactones, terpenoids, alkaloids, antibiotics, and metal-chelating agents [78]. Mechanism of insulin resistance using mushroom is as follows.

### 3.1. Blood Glucose-Lowering Effect of Polysaccharide

Polysaccharides are ubiquitous biopolymers made up of simple sugar or monosaccharides linked together by glycosidic linkage. Based on structure, polysaccharides are divided into two groups-homopolysaccharides (linear or highly branched, composed of the same monosaccharide molecules) and heteropolysaccharides (made up of different monosaccharide units) [79]. Earlier studies show that mushrooms are rich in *ß*-D-glucans, *ß*-glucan, a type of dietary fiber that shows promising health benefits against T2DM [80]. Mushroom extracts from *Pleurotus species* [56], *Boletu* [58], *Grifola frondosa* [81], *Agaricus bisporus* [38], *Hericium erinaceus* [63], and *Ganoderma lucidum* [82] regulate the synthesis of glycogen and lowers blood glucose levels by regulating gene expression of glycogen synthase kinase (GSK-3 *ß*), glycogen synthase (GS) and glucose transporter 4 (GLUT4) in liver and muscle. Therefore GSK-3 *ß* could be identified as an insulin-mediated GS-regulated negative regulator [83]. Other mechanisms by which polysaccharides prevent insulin resistance are by reducing α-amylase activity, α-glucosidase activities, and finally facilitating PI3K/AKT pathways, which are directly involved in glucose homeostasis [78].

Other mechanisms by which polysaccharides lowers blood glucose level is as follows-

#### 3.1.1. Inhibition of Glucose Absorption

Due to the presence of water-soluble dietary fiber, mushrooms delay glucose absorption and slow down digestion rates, thereby postprandial glucose upsurge [84,85]. Several studies have proven that mushrooms, especially *Pleurotus* spp. [56], *Grifola frondosa* [81], *Agaricus bisporus* [38], *Hericium erinaceus* [63], and *Ganoderma lucidum* [65] have significant blood glucose-lowering effect, as they delay the absorption of glucose and therefore improves hyperglycaemic condition [78].

#### 3.1.2. Maintains Pancreatic *ß* Cells Activity

Mushrooms polysaccharides (*ß*-D-glucan) act as a potent immune modulator and prevent activations of pro-inflammatory cytokines by reducing the activity of NF-kB, and it also outlawed oxidative damage. Bioactive compounds from mushrooms, especially polysaccharides, prevent pancreatic *ß*-cell apoptosis and hinder glucotoxicity [80]. Studies also showed that mushroom extracts from *Pleurotus* spp., *Boletus, Agaricus bisporus,* and *Hericium erinaceus* have a significant effect on *ß*-cell functionality and thus maintain *ß*-cell growth [63].

### 3.2. Blood Glucose-Lowering Effect of Terpenoids

Enzymes like α-glucosidase and α-amylase hydrolyze oligosaccharides to monosaccharides and increase blood glucose levels [86]. Mushrooms terpenoids (monoterpenes, diterpenes, sesquiterpenes, and triterpenes) (Figure 3) [19] from *Pleurotus* spp. [56], *Laetiporus sulphurous* [87], *Tremella fuciformis* [88], *Ganoderma lucidum* [60], and *Pholiota microspore* [89] are believed to have an α-glucosidase inhibitory activity that prevents the formation of monosaccharide molecules and facilitates glycogen formation in the liver and muscle.

### 3.3. Role of Vitamin D in Blood Glucose Regulations

Mushroom resides from the fungal kingdom, and unlike a plant, it has a high concentration of ergosterol in the cell wall. In the presence of sunlight, ergosterol in the mushroom cell wall is transformed to pre-vitamin D2 and thermally isomerized to ergocalciferol, commonly known as vitamin D2 [90]. 1, 5 (OH) 2D or 1,25-dihydroxy vitamin D plays an important role in glucose homeostasis. It also protects *ß*-cells from harmful immune attacks by its direct action on *ß*-cells, and indirect action on different immune cells, including inflammatory macrophages, dendritic cells, and a variety of T cells [91]. Molecular mechanisms by which vitamin D maintains insulin secretion is by regulating intracellular calcium concentration. Vitamin D, with the help of calbindin, facilitates Ca+ absorption, PKA activation, and PLC synthesis, which facilitates calcium secretion that, in consequence, leads to insulin secretion [92] (Figure 4).

Subsequently, the genomic mechanism of vitamin D action is mediated by the vitamin D receptor (VDR). The active form of vitamin D 1,25 (OH)2 D3 binds to VDR and forms a heterodimer with the retinoid receptors (RXR). The complex of 1,25 (OH)2 D3-VDR-RXR is translocated to the nucleus and binds to vitamin D-responsive elements (VDRE), thereby facilitating epigenetic modulations and preventing insulin resistance [93].

In short, vitamin D shows its function in different ways, such as inherited gene polymorphism, immune-regulatory functions, proliferation preventive actions, anti-inflammatory activities (decreases the functions of pro-inflammatory cytokines, TNF-α, IL-8b, and IL-6), and finally regulates the production of adipokines and thereby prevents insulin resistance (via IRS, AKT, PPARy, and VDRE gene regulations) [94] (Table 3).

The bioavailability of vitamin D in diabetic treatment is still controversial, and based on the available data, it is not yet transparent among scientists [95]. However, recent findings from randomized placebo trial data by Urbain et al., 2011 shows that button mushroom treated with UV-B can improve vitamin D2 bioavailability among human subjects, and the significant value does not differ with vitamin D2 supplements [96]. In addition, studies by Wenclewska et al., 2019; Mutt et al., 2020; Hajj et al., 2020; and Cojic et al., 2021 demonstrated the positive impact of vitamin D supplementation (2000IU and 30,000IU) on T2DM, which requires further trials [97,98,99,100]. Therefore, the question remains the exact mechanism and dose-dependent action of vitamin D on diabetes mellitus treatment.

## 4. Conclusions

In the present review, a total of 23 edible mushrooms have been scrutinized to review the medicinal components and diabetes mellitus preventive activities. Among all the varieties, 13 varieties have anti-diabetic properties. Mushrooms’ anti-diabetic activity is generally dependent on their polysaccharide (*ß*-D-glucan) and vitamin D contents. Therefore, in vivo and in vitro studies show that among 13 varieties, only 11 verities demonstrated anti-diabetic activities. Additionally, the most widely studied variations are *Pleurotus, Grifola,* and *Ganoderma* species. Thus, based on the available data, it can be concluded that mushrooms are beneficial fungi that have a great potential to treat non-communicable diseases, especially diabetes. However, future research work is necessary for the clinical field such as animal study (in vivo and in vitro), enzyme inhibition assay (α-amylase, α-glucosidase, pancreatic lipase, and DPP4-dipeptidyl peptidase 4), human trial, pilot study, as well as prospective and retrospective studies to use the possible therapeutic applications of mushrooms. Special focus must be given to the link between vitamin D and insulin resistance along with enzymatic assay by considering its potential effect. Furthermore, without adequate investigation, the conclusion is relatively challenging. So, further revelation is recommended in the clinical probe.

## Figures and Tables

**Figure 1 pharmaceutics-14-00436-f001:**
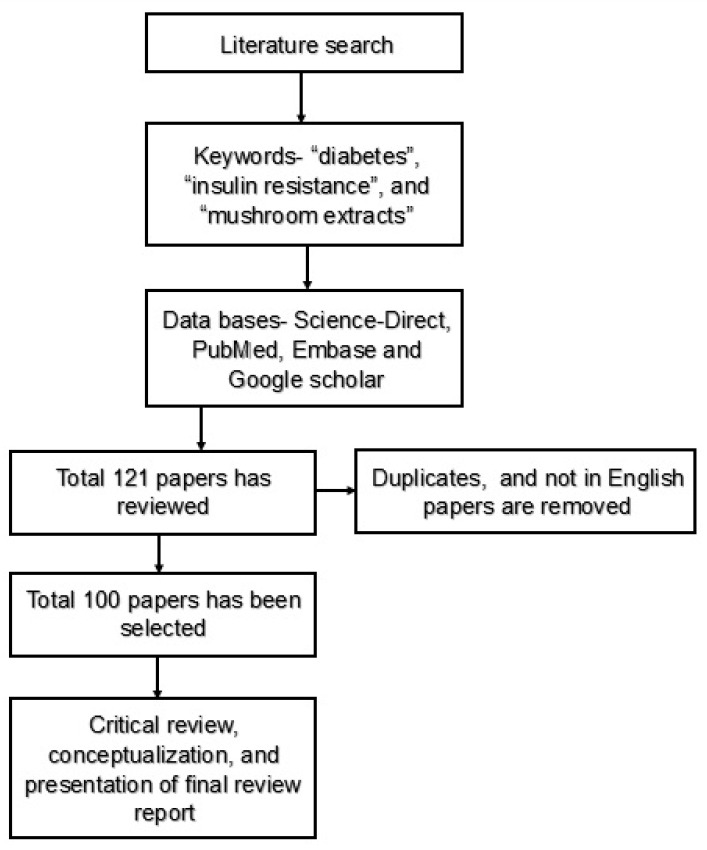
Review research methodology.

**Figure 2 pharmaceutics-14-00436-f002:**
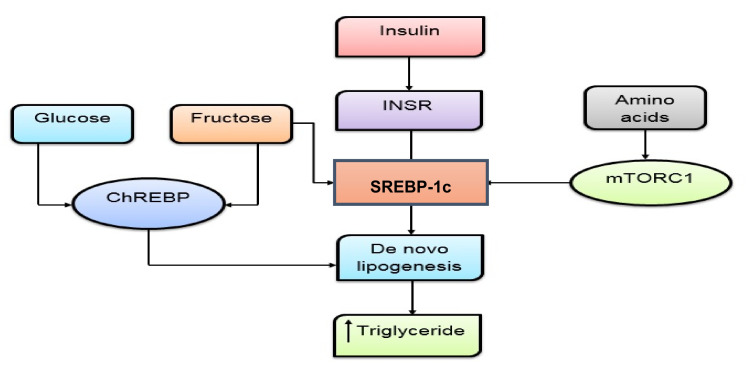
Relationship between liver insulin resistance and macronutrients. ChREBP—carbohydrate response element-binding protein, INSR—insulin receptor, SREBP-1c—sterol-regulated element-binding protein, mTORC1—mammalian target of rapamycin complex 1 or mechanistic target of rapamycin complex 1.

**Figure 3 pharmaceutics-14-00436-f003:**
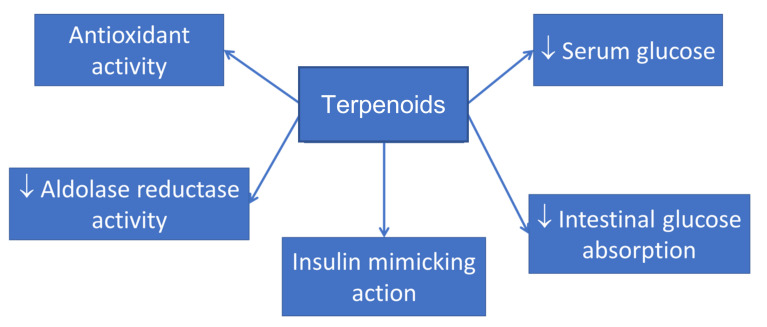
Beneficial effects of mushroom terpenoids (monoterpenes, diterpenes, sesquiterpenes, and triterpenes).

**Figure 4 pharmaceutics-14-00436-f004:**
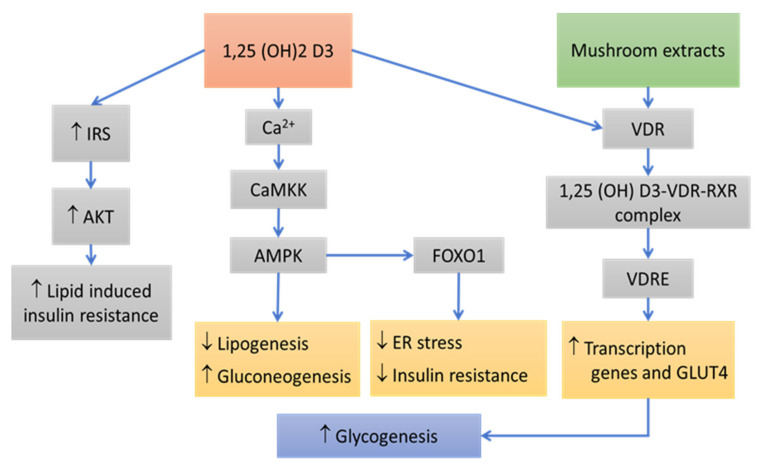
Relationship between vitamin D, mushroom extracts, and insulin resistance. IRS—insulin receptor substrate, AKT—protein kinase B, CaMKK—calcium/calmodulin-dependent protein kinase, AMPK—activated protein kinase, FOXO1—forkhead box transcription factor 1, VDR-, VDRE—vitamin D-responsive elements.

**Table 1 pharmaceutics-14-00436-t001:** Types of edible mushrooms.

S. No.	Scientific Name	Vernacular Name	Photos	Reference
1	*Craterellus aureus*	Cantharellus, chanterelles	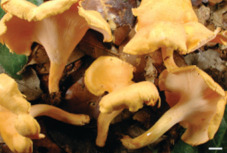	[31]
2	*Calvatia rugosa*	Puffballs	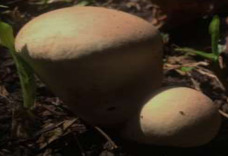	[32]
3	*Coprinus comatus*	Shaggy mane	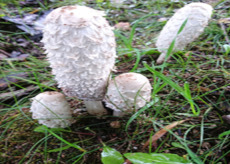	[33]
4	*Pleurotus ostreatus*	Oyster mushroom	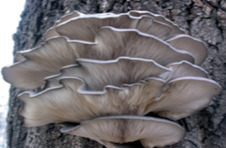	[34]
5	*Boletaceae Boletales*	Boletes	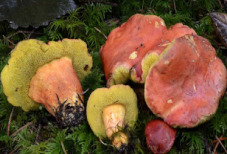	[35]
6	*Laetiporus sulphurous*	Sulfer shelf	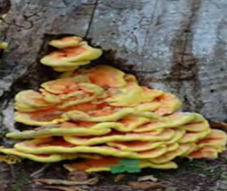	[36]
7	*Grifora frondosa*	Hen of the woods	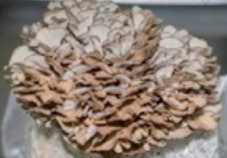	[37]
8	*Agaricus bisporus*	Button mushroom	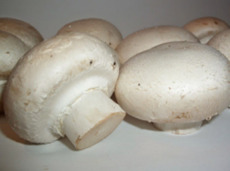	[38]
9	*Ramariopsis subarctica*	Coral fungi	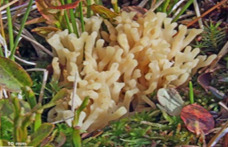	[39]
10	*Morachella esculenta*	Morels	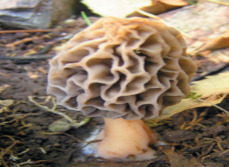	[40]
11	*Hericium erinaceus*	Bearded tooth	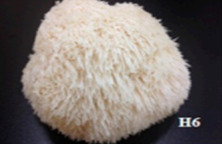	[41]
12	*Volvariella volvacea*	Straw mushroom	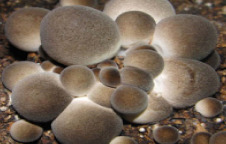	[42]
13	*Ganoderma lucidium*	Lingzhi mushroom	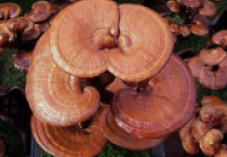	[43]
14	*Tremella fuciformis*	Snow fungus	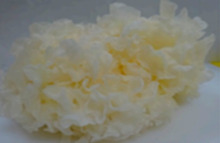	[44]
15	*Lentinus concentricus*	-	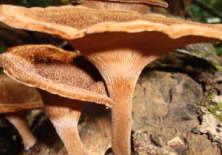	[45]
16	*Calocybe indica*	Milky white mushroom	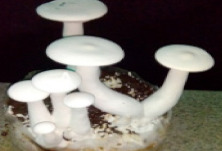	[46]
17	*Lenzites betulina*	Wood-rooting fungi	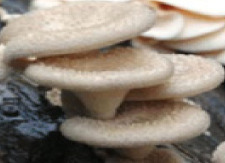	[47]
18	*Pholiota microspora*	Slippery mushroom	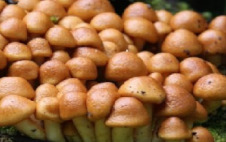	[48]
19	*Flammulina velutipes*	Enoki mushroom	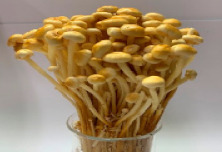	[49]
20	*Lentinula edodes*	Shiitake mushroom	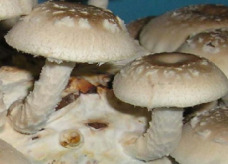	[50]
21	*Hypsizygus tessellatus*	Buna shimeji	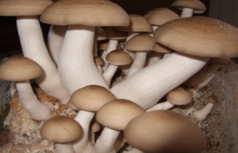	[51]
22	*Agrocybe aegerita*	Poplar mushroom, Chestnut mushroom, Velvet pioppini	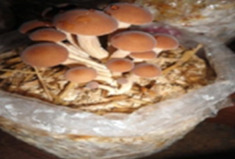	[52]
23	*Termitomyces robustus*	Termitomyces mushrooms	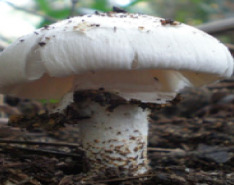	[53]

**Table 2 pharmaceutics-14-00436-t002:** Bioactive compounds of edible mushrooms and their effect on in vivo and in vitro study model.

S. No.	Scientific Name	Compounds	Functions	Models	Mushroom Doses	Mechanism/Action	Reference
1	*Calvatia gigantea*	2-Pyrrolidinone, 1-Dodecene, ergosterol, hexadecane, benzeneacetic acid.	Anti-diabetic, antioxidant, anti-inflammatory.	Alloxan-induced diabetic rat.	100, 200, and 400 mg/kg BW/day.	-Alpha-amylase inhibitory activity.	[54]
2	*Coprinus comatus*	Mycelium, polysaccharides.	Immunomodulatory, anti-diabetic, antioxidant, anti-cancer.	High-fat diet and STZ-induced mice.	400 mg/kg BW/day.	-Reduce BG level, relieves oxidative stress, ameliorate DN via PI3K/Akt and Wnt-1/β-catenin pathways.	[55]
3	*Pleurotus ostreatus, Pleurotus pulmonarius, and Pleurotus fossulatus*	Terpenoids, heterocyclic amines, phenols, glucan, proteoglycan.	Anti-cholesterol, anti-cancer effect, anti-inflammatory, anti-diabetic.	STZ and metformin-induced rat.	5–10% powder, 50, 80, 200, 250, 400, and 500 mg/kg BW/day extract.	-Decreases serum glucose level, alpha-amylase activity; -Increases P-AMPK and GLUT4 in muscle and adipose tissue; -Improves liver functions and maintains AST, ALT, and ALP levels.	[56,57]
4	*Boletus*	Tocopherol, quinic acid, hydroxy benzoic acid.	Antioxidant, anti-inflammatory, hypoglycemic.	STZ-induced rat.	400 mg extract/kg BW/day.	-Decreases TC, TG, TNF-alpha, and NF-kB level; -Maintains MDA level; -Improves antioxidant (CAT, SOD, and GSH) and CYP7A1 levels.	[58]
5	*Grifora frondosa*	Grifolan polysaccharide, D-fraction, MD-fraction, polysaccharide, galactomannan, heteroglycan.	Hypoglycemic, anti-inflammatory, anti-modulatory, anti-tumor.	STZ, alloxan-induced rat and palmitate-induced C2C12 cells.	0.5–20 µM (introduced to C2C12 cell), 112.5, 200, and 675 mg/kg BW/day extract (introduced to STZ- and alloxan-induced rat).	-Inhibits serum levels of IL-2, IL-6; -Modulates serum level of oxidant factors such as superoxide dismutase, glutathione peroxidase, catalase, malondialdehyde, and reactive oxygen species; -Increases glucose uptake and decreases ROS formation and up-regulates IRS-1, p-IRS-1, PI3K, Akt, pAkt and GLUT4 protein, and down-regulates p-JNK and p-p38 expression; -Improves insulin resistance and gut microflora content.	[59,60]
6	*Agricus bisporus*	Pyrogallol, hydroxybenzoic acid derivatives glavonoid.	Anti-inflammatory, anti-diabetic.	Alloxan-induced rat.	15–30 g/day, 250, 500 and 750 mg/kg BW/day.	-Improves antioxidant (SOD) level; -Improves ALP, AST, ALT level; -Reduces hyperlipidemia.	[38,61]
7	*Morachella esculenta*	Polysaccharides (mannose, galactose, and glucose), phenolic compounds.	Antioxidant, anti-inflammation, immunoregulation, hypoglycemic.	-	-	-	[62]
8	*Hericium erinaceus*	4-chloro-3,5-dimethoxybenzoic acid-O-arabitol ester, 2-hydroxymethyl-5-α-hydroxy-ethyl-γ-pyranone, 6-methyl-2,5-dihydroxymethyl-γ-pyranone, 4-chloro-3,5-dihydroxybenzaldehyde, 4-chloro-3,5-dihydroxybenzyl alcohol.	Immunomodulatory, hypoglycemic, antimicrobial.	STZ-treated rat.	400–600 mg/kg BW/day.	-Reduces blood glucose, BUN, and CRT level; -Maintains ALP, ALT, and AST levels; -Improves antioxidant level (SOD, glutathione).	[63,64]
9	*Ganoderma lucidium*	Ganoderic acid, danoderiol, danderenic acid, lucidenic acid, ganoderma leucidum polysaccharide.	Anti-diabetic, anti-inflammatory.	Metformin-, STZ-, and high-fat-treated rat.	1–3% freeze-dried mushroom, 25, 50, 100, 250, 500 and 1000 mg/kg BW/day extract.	-Decreases HBA1c and improves AST, ALT level.	[60,65]
10	*Lenzites betulina*	α-glucan, *ß*-glucan, *ß*-glucan protein, galacturonic acid.	Antioxidant, anti-hyperglycaemic, anti-inflammatory, anti-proliferative, antibacterial.	-	-	-	[66]
11	*Flammulina velutipes*	Flammulinolide, enokipodin, proflamin, other polysaccharide.	Anti-tumor, anti-hypertension, anti-hypercholesterolemia, hypoglycemic.	STZ-induced mice.	400 mg/kg, 600 mg/kg, and 800 mg/kg BW/day.	-Improves PI3K/AKT pathway.	[67,68]
12	*Lentinula edodes*	Lentinan, eritadenina.	Anti-carcinogenic, antioxidant, hypocholesterolemic action.	STZ-treated rat (gestational diabetes).	100 mg/kg BW/day.	-Improves maternal insulin level; -Reduces aminotransferase, aspartate aminotransferase, triglyceride, and total cholesterol level.	[69,70]
13	*Termitomyces robustus*	γ glutamyl-*ß*-phenylethylamine, tryptophan 1,4-hydroxyphenylacetic acid, hydroxyphenyl propionic acid and phenyllactic acid.	Hypoglycemic effect.	In vitro assay, Wister rat.	Crude extract 78.05 and 86.10 µg/mL. 500, 1000, and 1500 mg/kg BW/day. In vitro assay, rate acute toxicity test (10 g/kg extract) and subacute toxicity test (500, 1000, and 1500 mg/kg) BW/day.	-α-glucosidase and α-amylase inhibitory activity.	[15,71]

DN—diabetic nephropathy, Wnt-1—wingless-related integration site, P-AMPK—activated protein kinase, GLUT4—glucose transporter type 4, TC—total cholesterol, STZ—streptozotocin, FBG—fasting blood glucose, BGL—blood glucose level, TG—triglyceride, CRT—creatinine, BUN—blood urea nitrogen, AST—aspartate aminotransferase, ALT—alanine aminotransferase, ALP—alkaline phosphatase, TNF—alpha-tumor necrosis factor alpha, NF-kB—nuclear factor kappa B, MDA—malondialdehyde, CAT—chloramphenicol acetyltransferase, SOD—superoxide dismutase, GSH—glutathione, CYP7A1—cholesterol 7-alpha-monooxygenase gene, IL-6—interleukin 6, IL-2—interleukin 2, IRS-1—insulin receptor substrate 1, PI3K—phosphoinositide-3-kinase, Akt—protein kinase B, p-JNK-c—Jun N-terminal kinase, HBA1c—glycosylated hemoglobin, PGL—plasma glucose level.

**Table 3 pharmaceutics-14-00436-t003:** Functions of vitamin D on different organs [94].

Organ Name	Functions
Pancreas	Increases insulin secretion and enhances the transformation of pro-insulin to insulin
Skeletal muscle	Through VDR expression maintains glucose homeostasis
Skin	Improves skin micro-circulations and fasten wound healing
Nervous system	Improves nerve conduction and shows the analgesic effect
Kidney	Controls urinary albuminuria
Retina	Defend against oxidative stress

## Data Availability

The data that support the findings of this study are available from the corresponding author upon reasonable request.

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
