# Peer review of "Medicinal Components in Edible Mushrooms on Diabetes Mellitus Treatment"

_pharmaceutics, 2022, doi:10.3390/pharmaceutics14020436_

Round 1

Reviewer 1 Report

The manuscript review entitled "Medicinal components in edible mushrooms on Diabetes Mellitus treatment" from Das et al involved the comprehensive review analyses of edible mushrooms (extracts) role related to insulin resistance (bioactive compounds composition).

The introduction is according to the developed topic of the manuscript.

As a suggestion it would be important to analyze/check the additional updated published data regarding the topic of interest, such as:

Role of edible mushroom as a potent therapeutics for the diabetes and obesity - doi: 10.1007/s13205-019-1982-3

Edible Mushrooms: Novel Medicinal Agents to Combat Metabolic Syndrome and Associated Diseases - DOI: 10.2174/1381612826666200831151316

Therapeutic potential of mushrooms in diabetes mellitus: Role of polysaccharides - DOI: 10.1016/j.ijbiomac.2020.07.145

Also, the manuscript is organized, clear, and focused on the topic that is of growing interest due to the potential applications of natural products extracts (mushrooms).

Additionally, the mushroom information the authors described is supported with clear and logical images and tables that summarize all the reported data.

Furthermore, I encourage the authors to check some mistakes such as:

  • Please remember that in vitro and in vivo should be in italics
  • Figure 1 has the incorrect caption (the figure is related to terpenes not to polysaccharides), please double check and correct the caption and it is not specified in the main manuscript.
  • It would be important to identify which are the most important terpenoids (names) and polysaccharides included in the reported mushrooms in each paragraph (apart from Table 2).

I would like to invite the authors to add the abbreviation list of words at the end of this manuscript.

I recommend the acceptance of this manuscript after the authors performed the suggested corrections/additions.

Author Response

Response to Reviewer #1:

The manuscript review entitled "Medicinal components in edible mushrooms on Diabetes Mellitus treatment" from Das et al involved the comprehensive review analyses of edible mushrooms (extracts) role related to insulin resistance (bioactive compounds composition).The introduction is according to the developed topic of the manuscript. As a suggestion it would be important to analyze/check the additional updated published data regarding the topic of interest, such as:

-Role of edible mushroom as a potent therapeutics for the diabetes and obesity - doi: 10.1007/s13205-019-1982-3

 -Edible Mushrooms: Novel Medicinal Agents to Combat Metabolic Syndrome and Associated Diseases - DOI: 10.2174/1381612826666200831151316

 -Therapeutic potential of mushrooms in diabetes mellitus: Role of polysaccharides - DOI: 10.1016/j.ijbiomac.2020.07.145

Response: We thank the reviewer for the constructive suggestion. As recommended by the reviewer three papers by Tung et al., 2020; Dubey et al., 2019; and Khursheed et al., 2020 has included in the revised manuscript. Please refer lines 59, 173 and 188/ references - 17, 80, and 86)

  1. Tung, Y.T.; Pan, C.H.; Chien, Y.W.; Huang, H.Y. Edible Mushrooms: Novel Medicinal Agents to Combat Metabolic Syndrome and Associated Diseases. Curr Pharm Des 2020, 26, 4970-4981, doi:10.2174/1381612826666200831151316.
  2. Dubey, S.K.; Chaturvedi, V.K.; Mishra, D.; Bajpeyee, A.; Tiwari, A.; Singh, M.P. Role of edible mushroom as a potent therapeutics for the diabetes and obesity. 3 Biotech 2019, 9, 450, doi:10.1007/s13205-019-1982-3.

3. Khursheed, R.; Singh, S.K.; Wadhwa, S.; Gulati, M.; Awasthi, A. Therapeutic potential of mushrooms in diabetes mellitus: Role of polysaccharides. International Journal of Biological Macromolecules 2020, 164, 1194-1205, doi:https://doi.org/10.1016/j.ijbiomac.2020.07.145.

-Also, the manuscript is organized, clear, and focused on the topic that is of growing interest due to the potential applications of natural products extracts (mushrooms).

Response: Thank you for the useful comment.

-Additionally, the mushroom information the authors described is supported with clear and logical images and tables that summarize all the reported data.

Response: Thank you so much for the valuable response.

-Furthermore, I encourage the authors to check some mistakes such as:

Please remember that in-vitro and in-vivo should be in italics

Response: Thank you for informing us. Now the words has changed into italics (lines 96, 97, 121, 122, 260 and 265) please refer the revised manuscript.

-Figure 1 has the incorrect caption (the figure is related to terpenes not to polysaccharides), please double check and correct the caption and it is not specified in the main manuscript.

Response: Thank you for the comment. As suggested, Figure 1 is now corrected and represented at page 14, and number has changed to Figure 3. Now it is specified in the main manuscript. (Please refer page 2, line 61 and page 12, line 165)

-It would be important to identify which are the most important terpenoids (names) and polysaccharides included in the reported mushrooms in each paragraph (apart from Table 2).

Response: Thank you for the helpful comment. As per the suggestion important terpenoids and polysaccharides are now mentioned in revised manuscript. (Please refer page 13 and 14, lines 172, 193, 203 and 210)

-I would like to invite the authors to add the abbreviation list of words at the end of this manuscript.

Response: Thank you so much for the helpful remark. As recommended by the reviewer, list of abbreviations are now added in the revised manuscript (please refer page 16, line 284).

-I recommend the acceptance of this manuscript after the authors performed the suggested corrections/additions.

Response: Thank you so much for the valuable suggestions and recommendation.

Reviewer 2 Report

In this manuscript the authors the authors describe the possible use of edible mushrooms in terms of controlling issues related to diabetes and insulin resistance. Overall this is an interesting, if not very novel, topic. There are a few issues the authors should consider in a revision.

-The text must be heavily revised in terms of English. Besides grammar mistakes there are many issues that greatly affect understanding the underlying concepts.

-As is usual in this kind of paper the authors should include a flow chart showing in more detail and in graphical form how they searched and selected for the papers they discussed in terms of inclusion/exclusion criteria.

-As noted this is not a novel topic.  As can be seen in the link below just Reviews on Mushrooms and Diabetes reveal a long list of recent papers. The authors should at least look at some of these more recent papers and mention what unique perspective their own efforts bring to the field.

https://pubmed.ncbi.nlm.nih.gov/?term=Mushrooms+AND+Diabetes&filter=pubt.review&sort=pubdate

-In that regard the authors should discuss more thoroughly the possible pathways involved and their effects, as is the case in many other reviews. Most of paper is merely descriptive. The section on vitamin D is interesting, but the link with diabetes is doubtful.

-It is also interesting that no use of mushrooms in traditional medicines is mentioned. Shouldn't this be the case?

-The Conclusion section is also very limited and should include also a suggestion from the authors on Future Studies. What kind of studies do the authors recommend? In which models? With what species of mushrooms or compounds? What could be the priority in order to further the field in a logical and useful manner?

Author Response

In this manuscript the authors the authors describe the possible use of edible mushrooms in terms of controlling issues related to diabetes and insulin resistance. Overall this is an interesting, if not very novel, topic. There are a few issues the authors should consider in a revision.

-The text must be heavily revised in terms of English. Besides grammar mistakes there are many issues that greatly affect understanding the underlying concepts.

Response: As suggested by the reviewer, we carefully reviewed the manuscript point-by-point by native professional speaker and corrected English grammar in the revised manuscript. Moreover, procedures are now properly described in the revised manuscript (please refer to revised manuscript)

-As is usual in this kind of paper the authors should include a flow chart showing in more detail and in graphical form how they searched and selected for the papers they discussed in terms of inclusion/exclusion criteria.

Response: Thank you so much for the useful suggestion. According to the suggestion paper selection criteria has updated as flow chart in revised manuscript (Please refer Figure 1, page 3).

-As noted this is not a novel topic.  As can be seen in the link below just Reviews on Mushrooms and Diabetes reveal a long list of recent papers. The authors should at least look at some of these more recent papers and mention what unique perspective their own efforts bring to the field.

https://pubmed.ncbi.nlm.nih.gov/?term=Mushrooms+AND+Diabetes&filter=pubt.review&sort=pubdate

Response: Thank you for the suggestive comment. As per the recommendation citation has added and also mentioned the unique perspectives to prevent insulin resistance in revised manuscript. In case of mushrooms unique properties we did not found much relevant papers which can support our hypothesis. Furthermore, we are carrying out research work on edible mushrooms and we erected sufficient preliminary data to prove the unique prospective (unpublished data). As a review paper, we just focused on insulin resistance, and effect of mushroom extracts on expected positive outcomes. (Please refer page 2, lines 63 and 90-93; page 13, line 203, and citation 19).

  1. Dasgupta, A.; Acharya, K. Mushrooms: an emerging resource for therapeutic terpenoids. 3 Biotech 2019, 9, 369, doi:10.1007/s13205-019-1906-2.

-In that regard the authors should discuss more thoroughly the possible pathways involved and their effects, as is the case in many other reviews. Most of paper is merely descriptive. The section on vitamin D is interesting, but the link with diabetes is doubtful.

Response: Thank you for the suggestion. By considering the suggestion possible pathways are added (Figures 2 and 4,) and linked with vitamin D effectiveness. As mentioned the role of vitamin D is doubtful, yes it is, for which we believe that the need of future in-vitro and/or in-vivo and clinical probe. For instance studies by Wenclewska et al., 2019; Mutt et al., 2020; Hajj et al., 2020; and Cojic et al., 2021 demonstrated positive impact of vitamin D supplementation (2000IU and 30,000IU) on T2DM, which requires further trials. Also, the bioavailability of mushroom vitamin D2 has proven by Urbain et al., 2011.  So, the main focus goes to pathophysiology of action, mechanism, and effectiveness (dose dependent) of the research probe.

  1. Wenclewska, S.; Szymczak-Pajor, I.; Drzewoski, J.; Bunk, M.; Śliwińska, A. Vitamin D Supplementation Reduces Both Oxidative DNA Damage and Insulin Resistance in the Elderly with Metabolic Disorders. Int J Mol Sci. 2019, 20, doi:10.3390/ijms20122891.
  2. Mutt, S.J.; Raza, G.S.; Mäkinen, M.J.; Keinänen-Kiukaanniemi, S.; Järvelin, M.R.; Herzig, K.H. Vitamin D Deficiency Induces Insulin Resistance and Re-Supplementation Attenuates Hepatic Glucose Output via the PI3K-AKT-FOXO1 Mediated Pathway. Mol Nutr Food Res 2020, 64, e1900728, doi:10.1002/mnfr.201900728.
  3. El Hajj, C.; Walrand, S.; Helou, M.; Yammine, K. Effect of Vitamin D Supplementation on Inflammatory Markers in Non-Obese Lebanese Patients with Type 2 Diabetes: A Randomized Controlled Trial. Nutrients 2020, 12, doi:10.3390/nu12072033.
  4. Cojic, M.; Kocic, R.; Klisic, A.; Kocic, G. The Effects of Vitamin D Supplementation on Metabolic and Oxidative Stress Markers in Patients With Type 2 Diabetes: A 6-Month Follow Up Randomized Controlled Study. Front. Endocrinol 2021, 12, doi:10.3389/fendo.2021.610893.

-It is also interesting that no use of mushrooms in traditional medicines is mentioned. Shouldn't this be the case?

Response: Thank you so much for this interesting comment. In this review paper we keep focused on medicinal edible mushrooms. As we are from Nutrition Department we gave special focus on edible mushrooms to evaluate its functional properties (Anti-diabetic). For more clarifications please refer table 2, page - 8.

-The Conclusion section is also very limited and should include also a suggestion from the authors on Future Studies. What kind of studies do the authors recommend? In which models? With what species of mushrooms or compounds? What could be the priority in order to further the field in a logical and useful manner?

Response: Thank you for the useful comment. As suggested, conclusion part is updated with suggestions and future recommendations.

We would like to suggest future studies related to clinical trial like animal study (in-vivo and in-vitro), enzyme inhibition assay (α- amylase, α- glucosidase, pancreatic lipase and DPP4- dipeptidyl peptidase 4), human trial, pilot study, as well as prospective and retrospective studies to utilize the possible therapeutic applications of mushrooms. Among mushrooms except Pleurotus, Grifola, and Ganoderma, other edible species must be focussed. Furthermore, special attention must be given to the link between vitamin D and insulin resistance along with enzymatic assay, by considering its potential effect. (For more detail please refer the conclusion, page 15, line- 256) 

Round 2

Reviewer 2 Report

The authors have addressed my main concerns adequately. I have no further comments.